# The Application of Mesenchymal Stromal Cells and Their Homing Capabilities to Regenerate the Intervertebral Disc

**DOI:** 10.3390/ijms22073519

**Published:** 2021-03-29

**Authors:** Andreas S. Croft, Svenja Illien-Jünger, Sibylle Grad, Julien Guerrero, Sebastian Wangler, Benjamin Gantenbein

**Affiliations:** 1Tissue Engineering for Orthopaedics & Mechanobiology (TOM), The Department for BioMedical Research (DBMR) of the Faculty of Medicine of the University of Bern, University of Bern, CH-3008 Bern, Switzerland; andreas.croft@dbmr.unibe.ch (A.S.C.); julien.guerrero@dbmr.unibe.ch (J.G.); 2Department of Orthopaedics, Emory University School of Medicine, VAMC, 1670 Clairmont Rd, Decatur, GA 30033, USA; svenja.illien-junger@emory.edu; 3AO Research Institute Davos, CH-7270 Davos, Switzerland; sibylle.grad@aofoundation.org; 4Department of Orthopaedic Surgery and Traumatology, Inselspital, Bern University Hospital, University of Bern, CH-3010, Bern, Switzerland; sebastian.wangler@insel.ch

**Keywords:** mesenchymal stromal cells, homing, cell-based therapy, intervertebral disc regeneration

## Abstract

Chronic low back pain (LBP) remains a challenging condition to treat, and especially to cure. If conservative treatment approaches fail, the current “gold standard” for intervertebral disc degeneration (IDD)-provoked back pain is spinal fusion. However, due to its invasive and destructive nature, the focus of orthopedic research related to the intervertebral disc (IVD) has shifted more towards cell-based therapeutic approaches. They aim to reduce or even reverse the degenerative cascade by mimicking the human body’s physiological healing system. The implementation of progenitor and/or stem cells and, in particular, the delivery of mesenchymal stromal cells (MSCs) has revealed significant potential to cure the degenerated/injured IVD. Over the past decade, many research groups have invested efforts to find ways to utilize these cells as efficiently and sustainably as possible. This narrative literature review presents a summary of achievements made with the application of MSCs for the regeneration of the IVD in recent years, including their preclinical and clinical applications. Moreover, this review presents state-of-the-art strategies on how the homing capabilities of MSCs can be utilized to repair damaged or degenerated IVDs, as well as their current limitations and future perspectives.

## 1. Introduction

### The Burden of Low Back Pain and Its Association with Intervertebral Disc Degeneration

Being the leading cause of disability worldwide, low back pain (LBP) provokes agony in up to 637 million people worldwide [1]; it accounts for more lost workdays than any other musculoskeletal disorder in the USA, and in Europe, LBP is the most prevalent reason for premature retirement and medically certified sick leave [2,3,4]. Even though LBP occurs predominantly among female individuals between the ages of 40 and 80 years, no age group is spared from LBP, and also countless children worldwide suffer from this condition [5]. In addition to being a personal struggle for each individual sufferer combined with a decreased quality of life, LBP is also responsible for a remarkable economic burden, causing expenses between $100 and $200 billion per year in the US alone [6]. Strikingly, two-thirds of these expenses are indirect costs, primarily due to lost wages [6]. Even though most back pain episodes start all of a sudden and usually are of short duration, they can also become chronic [3,7]. As a result, the burden of LBP remains long term or even a lifetime [8]. The causes of LBP are manifold; it can be idiopathic or initiated by vertebral fractures, infections, spinal tumors, or by intervertebral disc degeneration (IDD) [3,9]. In particular, the latter deserves attention, as IDD is the main contributor to LBP [10]. However, in order to understand the possible correlations between IDD and LBP, one should consider the general structure of a healthy intervertebral disc (IVD) and how a pathophysiological development of the IVD can potentially lead to this disease.

IVDs are the biggest avascular organs in the human body, and they are essential for the spine to move with six degrees of freedom [11,12]. Additionally, due to its unique architecture, an IVD is also an excellent shock distributor. This feature is mainly based on the IVD’s highly hydrated core, the nucleus pulposus (NP), which consists of up to 88% water [13]. The water is retained in the NP’s extracellular matrix (ECM) by negatively charged proteoglycans, which are structured by a network of collagen type II (COL2) fibers [14]. The NP is surrounded and held in place by the resilient annulus fibrosus (AF), a tissue rich in collagen type I (COL1) fibers organized in concentric lamellae [15]. Finally, the NP and AF are enclosed by two hyaline cartilaginous endplates (CEP) on superior and inferior sites [15]. A physiologically healthy IVD is characterized by a well-balanced microenvironment, which is defined by low oxygen levels, high mechanical stress, high osmotic pressure, and low pH [16].

Due to the IVD’s avascular nature, its cells depend on the blood vessels’ nutritional supply at the IVD margins [17]. Hence, a reduction in the supply of nutrients has been correlated with IDD [18]. This cut of essential nutrients can be caused by blocked capillaries near the IVD or calcified CEPs, preventing the supplements from properly diffusing into the IVD [17]. Consequently, a lack of nutrients can be responsible for changes in the IVDs catabolic and anabolic turnover, cause dehydration, and lead to an aberrant production of pro-inflammatory cytokines [19,20]. As a result, these cytokines can initiate inflammation, promote vascular and nerve ingrowth into the IVD, induce pain, or even increase the risk of herniation (Figure 1) [20,21,22,23]. In addition to the lack of nutrient diffusion into the IVD, other risk factors such as excessive mechanical stress, smoking, trauma, and unfavorable genetic predispositions have been correlated with IDD [1]. In the past, multiple studies have confirmed that many signs of IDD, i.e., an anterior or posterior bulge of the IVD, an insufficient blood supply, or a decreased signal intensity of the NP in a T2-weighted image, are correlated with an increased risk of developing LBP [18,24]. Nevertheless, the exact cause of IDD and its relation to LBP are still under investigation.

There are many options to treat LBP, yet none of them come without any drawbacks or limitations. Acute or subacute conditions are usually treated non-invasively and preferably non-pharmacologically [25]. Recommended therapies include acupuncture [26,27], physical therapy [28], exercise or superficial heat [29]. Additionally, for patients who suffer from chronic LBP, non-pharmacological treatments are usually considered first [25]. If these treatment options fail to reduce symptoms, pharmacological therapies are considered. They include the application of non-steroidal anti-inflammatory drugs (NSAIDs), acetaminophen, or opioids [29]. Nonetheless, in the case of inadequate response to conservative therapies, more invasive interventions are often required. Currently, the “gold-standard” surgery to treat chronic LBP associated with IDD is a discectomy, followed by spinal fusion [30]. In brief, the degenerated IVD is removed, and the cavity between the adjacent vertebral bodies is replaced by a cage containing bone grafts or substitutes that can be supplemented with osteogenic inductive growth factors such as bone morphogenetic protein 2 (BMP2) to induce ossification [31,32]. Then, pedicle screws are inserted to improve the mechanical stability and to immobilize the adjacent vertebral bodies (Figure 2) [33]. 

Nonetheless, the efficacy of spinal fusion to reduce patient’s pain and disability is controversial. In the past, multiple studies have reported that spinal fusion is not more effective than non-surgical care for patients suffering from chronic LBP [34,35]. Therefore, the focus of ongoing IVD research has shifted increasingly towards regenerative approaches aiming to slow down or even reverse the ongoing degenerative cascade. The degenerative environment has to be stabilized to achieve this goal and eventually shifted towards the IVD’s physiological condition. Potential strategies include (i) blocking of pro-inflammatory cytokines/proteinases in the ECM, (ii) reverse the gene expression of pro-inflammatory cytokines/proteinases in the resident disc cells, or (iii) stimulating resident disc cells to produce novel ECM [36]. However, IVD degeneration is also associated with cell death resulting in a lack of regenerative capacity. Therefore, the application of additional cells into the degenerative environment has been introduced as a potential strategy to support regeneration [36]. A promising approach to induce IVD regeneration could be the use of progenitor cells such as the recently discovered NP cells that are positive for Tie2 marker (aka. angiopoietin-1 receptor) and disialoganglioside 2 (GD2) [37], as well as the use of stem cells such as induced pluripotent stem cells (iPSCs) [38]. Yet, the application of mesenchymal stromal cells (MSCs) has been studied most extensively with regard to IVD regeneration [36,39,40,41].

For this reason, the purpose of this narrative literature review is to show the achievements and progress made on MSC-based therapy with the eventual aim to regenerate damaged and/or degenerated IVDs. A particular focus of this review is targeted on the state-of-the-art strategies of MSC homing into IVDs, their potential, but also their current limitations for cell-based therapies.

## 2. Application of Mesenchymal Stromal Cells for IVD Repair

The sources of MSCs in the human body are manifold. In addition to the bone marrow, MSCs have been found in the cord blood, peripheral blood, adipose tissue, fetal tissue, and the liver [42]. It has been shown for many years now that MSCs have multipotent characteristics enabling them to differentiate towards lineages of mesenchymal tissues [43,44]. Moreover, MSCs are also believed to secrete a broad spectrum of bioactive factors to regulate the microenvironment at the site of action, referred to as “trophic activity” [45]. These factors can further stimulate cell proliferation of endogenous stem or progenitor cells, enhance the expression of ECM proteins, downregulate the expression of pro-inflammatory cytokines and inhibit apoptosis and scarring of the remaining tissue [45,46,47].

### 2.1. Preclinical Studies with MSCs

Due to these notable characteristics, research groups worldwide have tried to incorporate these features to regenerate the IVD (Table 1). Therefore, researchers either aim to differentiate MSCs into IVD-like cells and/or make use of MSC “trophic activity”. For example, Stoyanov et al. looked at approaches for directing bone marrow-derived MSCs into a NP-cell-like phenotype [48]. They demonstrated that MSCs cultured in a hypoxic environment and supplemented with growth/differentiation factor 5 (GDF5) responded with an upregulation of IVD ECM related genes (aggrecan (*ACAN*), *COL2*) and NP markers (cytokeratin 19 (*KRT19*), forkhead box F1 (*FOXF1*), carbonic anhydrase 12 (*CA12*)). To a lesser extent, this was also seen when co-culturing the MSCs with bovine NP cells. Consequently, they concluded that MSCs can acquire an NP-cell-like phenotype and can therefore be considered for IVD regenerative therapy. A similar approach to stimulate IVD-cell-like differentiation of MSCs with exogenously supplemented growth factors was documented by Clarke et al. [49]. They discovered that in addition to GDF5, transforming growth factor-beta 3 (TGF-β3) and in particular growth/differentiation factor 6 (GDF6) significantly increased the expression of the NP marker genes cytokeratin 8 (*KRT8*), cytokeratin 18 (*KRT18*), *KRT19*, *FOXF1,* and *CA12* as well as the production of glycosaminoglycans (GAG) in MSCs.

MSCs in combination with hydrogels or scaffolds have also been a popular option for tissue engineering strategies. Scaffolds ideally mimic the tissue and provide structural support for cell attachment and/or proliferation as well as ECM accumulation [63]. For instance, Frauchiger et al. [64] used genetically engineered silk fleeces functionalized either with GDF6 or TGF-β3 and seeded the silk with human bone marrow-derived MSCs. They demonstrated the high biocompatibility of silk for MSCs, whereby MSCs tended to differentiate towards an NP-like phenotype under these conditions. Moreover, many tissue engineering strategies for IVD repair include the combined usage of MSCs with hydrogels. For example, Peroglio et al. [65] investigated the differentiation of human MSCs, which were embedded into a thermo-reversible hyaluronan-based hydrogel. They showed that MSCs seeded in this hydrogel and subsequently delivered into a bovine IVD showed enhanced disc-like differentiation than respective cells that were first pre-differentiated with various growth factors and then implanted into the IVD. Furthermore, MSCs seeded in the hyaluronan hydrogel responded with a significant upregulation of *COL2*, *SOX9*, and a cluster of differentiation 24 (*CD24*) compared to the culture within alginate, indicating noticeable IVD-like differentiation of MSCs. 

A different kind of hydrogel has been tested by Zhang et. al, consisting of a combined triple interpenetrating network made of dextran, chitosan, and teleostean [66]. In combination, the three components form a stable hydrogel that mimics the mechanical properties of NP tissue. The objective of the in vivo study was to inject this hydrogel with and without the addition of MSCs into the NP of degenerated goat lumbar IVDs and to evaluate its therapeutic effect. Two weeks after treatment, a significant improvement in the IVD’s height was visible compared to the untreated controls as well as enhanced histological conditions. The authors state that the combined treatment with MSCs and hydrogel generally evoked a greater therapeutic effect than the hydrogel alone. Only tumor necrosis factor-alpha (TNF-α) expression levels were less favorable in the combined condition.

Recently, an ambitious study was conducted by Sun et al. with the approach of using an anatomically correct 3D printed IVD scaffold made of different biomaterials, incorporated growth factors, and MSCs to replace a defective IVD [67]. In brief, the core and NP analogue consisted of a hydrogel mixed with bone marrow-derived MSCs and TGF-β3. The AF’s printing material was composed of the same components as the NP, but connective tissue growth factor (CTGF) was added. To ensure sufficient mechanical support for the scaffold, a framework made of polycaprolactone (PCL) was printed. In vitro testing revealed high cell viability levels (up to 99%) after seven days of culture. Furthermore, MSCs in the scaffold’s “NP” core showed significantly upregulated levels of *ACAN* and *COL2* compared to controls, and the cells printed into the surrounding “AF” region expressed significantly more *COL1*, suggesting differentiation into AF-like cells.

### 2.2. Clinical Studies with MSCs

MSCs are very attractive from a clinical perspective as they can be isolated safely from the patient’s tissue and because of their negligible immunogenicity [68]. For these reasons, both autologous and allogeneic transplantations are enabled without the need for immunosuppressive drugs and with little ethical dispute [68]. In the past, a couple of clinical studies with predominantly positive outcomes have been carried out, in which MSCs were used to improve LBP associated with IDD [59,69]. In a pilot study by Orozco et al., ten patients diagnosed with lumbar disc degeneration and suffering from chronic LBP were injected with 10 million autologous bone marrow-derived MSCs per IVD [70]. Three months after the surgery, lumbar pain and disability were strongly reduced in 85% of the patients. Even after six and twelve months of follow-up, moderate improvements could be observed, including a significant increase in the IVD’s water content [70]. Nevertheless, the treatment did not manage to restore the IVD’s height. A recent randomized controlled study including 24 patients diagnosed with lumbar disc degeneration investigated the effect of intradiscal MSC injection. Patients were either treated with an injection of 25 million allogeneic bone marrow-derived MSCs per IVD or with a sham infiltration within the paravertebral musculature for the controls [71]. One year after surgery, the MSC-treated patients showed a significant reduction in pain in the lumbar region, reduced disability, and a decreased Pfirrmann grade. On the other hand, the control group did not show any amelioration of lumbar pain nor any improvements of disability, and the Pfirrmann grade even worsened significantly after one year. Favoring results regarding the treatment of culture-expanded MSCs were also found in a pilot study by Centeno et al. [72]. The study analyzed prospectively collected data up to seven years from 33 patients suffering from LBP and diagnosed with a posterior bulge of the IVD. All patients received treatment with an injection of autologous MSCs. As a result, LBP remained consistently lower than pretreatment, and for 85% of the patients, a reduction in the IVD bulge size was noticed. In addition, no safety issues such as infection, tumor growth, or death were recorded. Table 2 summarizes the published clinical studies related to intradiscal MSC transplantation.

Given the promising results observed with intradiscal cell injection, one can intuitively state that the best option for stem cell delivery into the IVD would be via needle injection. The main advantage is that a fixed number of cells can be precisely injected to the desired site of action. However, there are some considerable downsides. One problem of direct injection into the IVD is that the puncture causes trauma to the IVD to some extent and can therefore further aggravate its degenerated state [77,78]. Strikingly, multiple studies have used needle puncture as a model for degenerated IVDs [79,80]. An additional disadvantage of MSC injection is that the cavity created by the needle can enable the cells to reflux out of the IVD. Not only will this cause a loss of MSCs at the place where the regeneration or treatment would be desired, but escaped cells can also induce the adverse formation of osteophytes by altering the CEP tissue [81]. Furthermore, MSCs drawn from their native environment are suddenly exposed to the harsh environment of an IVD after injection, which leaves cells only a little time to adapt to these new surroundings. Although the survival rate of post-transplanted MSCs is uncertain, it is believed that most of these cells do not survive long term in a degenerated IVD. Consequently, this results in an accumulation of necrotic and apoptotic cell debris and thus may have detrimental effects on IVD homeostasis [82].

### 2.3. MSC Homing into an IVD

Because of the mentioned issues associated with direct cell injection, extensive efforts have been made to find alternative options for delivering MSCs into the IVD. In the last decade, great emphasis has been placed on the possibility of homing exogenous and endogenous MSCs into the IVD. Homing of MSCs is known as a process where cells are recruited from their initial niche to injured or pathological tissue [83]. Thereby, cells are mobilized into the peripheral bloodstream and migrate to the damaged tissue or organ [84,85]. Bone marrow-derived MSCs are known to be well capable of homing to numerous injured sites in the body such as myocardial infarction [86,87], traumatic brain injury [88], nephropathy [89], lung injury [90], bone fractures [91] and degenerated IVDs [92,93]. However, knowledge about the exact process and mechanism of how MSCs are mobilized and guided to the effector location is still incomplete. Nevertheless, it is strongly suspected that MSCs are navigated by multiple cell-signaling molecules and that the injured tissue itself expresses distinctive receptors or ligands to promote the infiltration of MSCs into the affected area [94]. To study this migration potential, Ponte et al. compared the in vitro migration capacity of bone marrow-derived MSCs through Transwell dishes under the influence of 16 growth factors and chemokines [95]. Before the migration assays were started, part of the cell population was preincubated either with TNF-α or interleukin-1β (IL-1β) to assess the impact of inflammatory cytokines on the migration capacity. They concluded that many growth factors and chemokines attracted a significant amount of unstimulated MSCs; though, growth factors were generally more efficient than chemokines, the most effective ones being insulin-like growth factor-1 (IGF-1) and platelet-derived growth factor-AB (PDGF-AB). Interestingly, cells that had initially been incubated with TNF-α increased their sensitivity for many chemokines, but only for one single growth factor tested. Here, the chemokines with the most potent effect on MSCs were RANTES (aka. CCL5), macrophage-derived chemokine (MDC), and stromal cell-derived factor-1 (SDF-1) [95]. In particular, SDF-1 (aka. CXCL-12) and its receptor C-X-C chemokine receptor type 4 (CXCR4) have been identified to play an essential role in homing of multiple cell types [96]. In a former study, an injectable hydrogel based on hyaluronan-poly(N-isopropylacrylamide) and supplemented with SDF-1 was used to recruit human MSCs in induced degenerated IVDs ex vivo [97]. The MSCs were applied onto the endplate of the organ cultured IVDs. The investigators showed that IVDs injected with SDF-1 together with the hydrogel not only attracted significantly more MSCs than IVDs treated with the hydrogel only, but the migration distance covered in the IVD was also significantly greater in these samples. Additionally, they also found that the migration of MSCs from younger donors was noticeably higher than the migrating distance of older ones.

In another study, Pattappa et al. performed a proteomic analysis to identify the chemoattractants released by degenerated IVDs, which would enable MSCs to home into the tissue [98]. Therefore, IVDs were cultured in a bioreactor that simulated either physiological or degenerative conditions for the IVDs (Figure 3). They found that RANTES, CXCL6, and IL-1β concentrations were elevated in the degenerated IVD’s media compared to the physiological conditions, whereby only the increase in RANTES was significant. Furthermore, a Boyden chamber assay was used to evaluate the chemoattractive properties of IVD-secreted RANTES. Here, depletion of RANTES in the IVD conditioned medium significantly decreased the number of migrated MSCs in the chamber and revealed that this chemokine is a key chemoattractant for MSCs in degenerated IVDs [98].

Based on the findings that RANTES could attract MSCs, Wangler et al. examined the subpopulation of cells that were mainly involved in the homing process in degenerated IVDs [99]. After characterizing the gene expression profile and the surface protein level of MSCs that migrated towards RANTES, they found that 60–90% of all migrated cells expressed the surface marker cluster of differentiation 146 (CD146). MSC subpopulations with CD146 as a surface marker were associated with a greater homing potential both in vitro and in organ culture using a bioreactor to either simulate physiological loading on IVDs or to induce its degeneration by applying high-frequency loading on the IVDs. Moreover, CD146-positive cells also showed an increased production of GAG when cells were cultured as pellets. Surprisingly, homing or injection of CD146-negative cells resulted in a higher sulphated GAG synthesis rate than CD146-positive cells in an IVD organ culture model [99].

Knowing that MSCs can be attracted by growth factors and chemoattractants, which are predominantly released by degenerated IVDs, the question that arises is: how efficiently are MSCs able to migrate into the affected IVD if they are released into the bloodstream from their initial storage location? Or in relation to cell therapy, how well do MSCs migrate to a degenerated IVD and contribute to its regeneration if they are injected intravenously? To address this question, Tam et al. compared the regeneration of degenerated IVDs following systemic or intradiscal MSC application [100]. Thereby, they used a mouse model in which IDD was induced by needle puncture, and multipotent stem cells derived from human umbilical cord blood to induce regeneration. First, they showed that the homing ability of intravenously injected cells was relatively limited, as cells were only found in the marrow space of the IVD and not in the NP, AF, or CEP. Moreover, only direct disc injection prevented a significant loss of the discs’ height. However, both delivery methods enhanced GAG production and upregulated the gene expression of *ACAN* relative to controls. The investigators suggested that these anabolic findings were likely due to paracrine effects caused by the MSCs [100]. Beneficial effects of systemically delivered MSCs were also found by Cunha et al. [101]. They assessed the impact of intravenously injected MSCs on a rat IVD lesion model. Based on IVD height index and histological grading, they concluded that transplanting MSCs 24 h after the initiation of an IVD injury leads to significantly less herniation and IDD than the injection of dermal fibroblasts, which were used as control. Furthermore, they highlighted the systemic immunomodulatory effect of injected MSCs represented by an upregulation of the cytokines IL-2, IL-4, IL-6, and IL-10 and a downregulation of IL-13 as well as TNF-α. Nonetheless, they dismissed the opportunity to track the injected cells and to confirm their presence at the damaged IVD. Consequently, this would have created a direct correlation between the homing properties of MSCs and their role in IVD regeneration. A similar approach was also carried out by Sakai et al. [92]. Here, they harvested bone marrow-derived green fluorescent protein (GFP)-expressing cells from mice and injected them into the tail vein of the host mice, which had been subjected to bone marrow irradiation and IDD. The investigators showed that only a limited number of injected cells migrated into the IVD, whereas most cells were found in the peripheral bone marrow and the CEP’s vascular canals. However, there was a clear relation between the severity of the induced IDD and the amount of recruited cells [92]. The data from Sakai et al. point out the difficulty of the cells to migrate into the IVD, as healthy IVDs are usually completely avascular, whereby the capillaries terminate at the CEPs and outermost AF [102,103]. Nevertheless, degenerated IVDs are also reported to induce vascular ingrowth in the outer layers of the AF [22]. Even though this neovascularization is seen as a pathological process in the IVD, it could potentially facilitate MSCs to reach the damaged sites and promote regeneration. However, this hypothesis has yet to be proven.

In order to identify how obstructive CEPs are for MSCs to home into the inner IVD, bovine IVDs with and without CEPs were cultured in a bioreactor system under physiological and pathological conditions in a study by Illien-Jünger et al. [93]. Interestingly, removal of the CEPs did not foster MSC migration, and MSCs were located mainly in the NP of degenerated IVDs. Moreover, Boyden chamber assays confirmed the chemoattractive properties of the culture medium, which was collected from degenerated IVDs. The higher the proportion of medium from degenerated IVDs, the greater the chemoattractive response of the MSCs.

The ability of stem cells to move through the endplate into the IVD and affect IVD metabolism was confirmed by Pereira et al. [104]. They used nucleotomized bovine IVDs and seeded human MSCs on the endplate surface. After up to three weeks of ex vivo culture, the group discovered that MSCs successfully migrated from the CEP into the inner IVD as far as the lesion area. Furthermore, cell migration significantly upregulated the expression of matrix proteins ACAN and COL2. Additionally, migration enhanced the production of growth factors, including fibroblast growth factor-6 (FGF-6) and 7 (FGF-7), insulin-like growth factor-1 receptor (IGF-1sR), and platelet-derived growth factor receptor (PDGF-R).

Another approach to investigate the influence of homed MSCs on degenerated IVDs was taken by Wangler et al. In brief, human MSCs were seeded on bovine whole IVDs and on either healthy, traumatic, or degenerated human IVD tissues [105]. After a period of MSC homing into the IVDs, they evaluated the cells’ effect on the host’s Tie2 progenitor cell populations, the IVD cell survival, and proliferation. The study showed that homed stem cells enhanced the proportion of Tie2-positive cells in both bovine and human IVD samples. Additionally, a proliferative response of the IVD cells was observed, and finally, MSC homing reduced the fraction of dead cells in the IVD. Table 3 summarizes the recent studies related to the homing capabilities of MSCs into the IVD.

However, to verify the potential of MSCs homing into a damaged or degenerated IVD and to assess the pathbreaking aspect of this innovative treatment for cell-based approaches to regenerate the IVD, preclinical in vivo studies on larger animals and clinical trials need to be performed. These trials will reveal how efficiently MSCs can be recruited from their source of production and storage, their stem cell niche within the IVD, or through intravenous cell injection. Furthermore, in the case of successful homing, the homed MSCs would have to prove their superiority against direct cell injection into the IVD. Only if the migrated cells can reduce LBP and promote IVD regeneration, clinical translation will be successful. Nevertheless, the advantage of MSC homing over direct injection into the IVD seems obvious, namely that no further damage to the IVD is being caused by needle puncture.

## 3. Limitations of MSC-Based IVD Regeneration

Despite the extensive number of studies and clinical trials that have been carried out in recent years, MSC-based cell therapy comes with some major drawbacks. As an example, there is an exponential decline of the MSC reservoir in the bone marrow with aging when autologous cell transplantation was considered [45]. A newborn has approximately one MSC per 10,000 bone marrow cells. As a teenager, the reservoir is already reduced to one MSC per 100,000 bone marrow cells, and as an elderly person aged approximately 80 years old, the fraction of MSCs in marrow cells decreases to 1 per 2 million [45]. As a matter of fact, this notable reduction in MSC titers creates a dilemma, as middle-aged and elderly people are usually the ones most affected by IDD and LBP, though they are usually the ones to show a relatively small number of MSCs in case of autologous cell therapy [106]. While this can be partially counteracted with ex vivo cell expansion, MSCs are known to lose their differential potential overtime with increasing cell population doubling levels and increasing age [107,108]. Furthermore, the inter-donor variability in MSC quality and quantity has to be considered [109]. Alternatively, the use of allogeneic cells from younger donors is a valuable option. In this case, however, the question arises of how easy and feasible it is to find enough potential donors. Another obstacle for successful cell therapy with MSCs is the previously described harsh microenvironment of an IVD [16]. It is yet unknown how well bone marrow-derived MSCs can adapt to this environment and survive over a long period. Nevertheless, hydrogels could be a promising solution to overcome this issue [110].

## 4. Conclusions

For years, spinal fusion has been considered the “gold standard” to surgically treat chronic discogenic LBP. However, because the technique intends to replace the IVD rather than regenerate the existing IVD, great efforts have been made to develop more biological therapies. Thereby, the application of MSC-based therapies has been placed into the spotlight. Many preclinical in vitro studies have been carried out to direct these cells into an IVD cell-like phenotype using exogenously added growth factors, thereby making the MSCs increasingly compatible and efficient for cell-based therapy. In ex vivo and in vivo studies, hydrogels and scaffolds transplanted with embedded MSCs have also shown promise. Even if the ideal biomaterial for IVD regeneration is yet to be found, novel biomaterials mimic the IVD tissue, providing adequate structural and biochemical support for MSCs. Finally, multiple clinical studies have reported efficient amelioration of LBP and a general improvement in the Pfirrmann grade following intradiscal MSC injection. However, long-term data are still missing that show how well injected bone marrow-derived MSCs can adapt to the harsh environment of an IVD and exhibit an actual regenerative effect.

In the past decade, alternatives to intradiscal cell injection have been sought. Probably the most promising approach seems to take advantage of MSC homing capabilities. In vitro cell and organ culture studies have shown that MSCs can be attracted by chemokines and growth factors released by degenerated IVDs, indicating that this strategy can potentially be used for IVD regeneration. Nevertheless, some controversies remain about the efficiency of exogenously administered MSCs to home in a degenerated IVD in vivo. While some studies claim that no or only a minimal number of administered MSCs can migrate into the IVD and therefore only have a limited regenerative potential, others suggest that already low numbers of migrated cells are sufficient to exert a regenerative effect. In general, the respective studies agree that homed MSCs positively affect degenerated IVDs, including immunomodulatory effects, an upregulation of anabolic genes, cell survival and proliferation, and the production of growth factors. Nonetheless, there is still a lack of preclinical in vivo studies on large animals, and ultimately clinical trials, on the ability of MSCs to home into degenerated or damaged IVDs and consequently their impact on IVD regeneration.

## Figures and Tables

**Figure 1 ijms-22-03519-f001:**
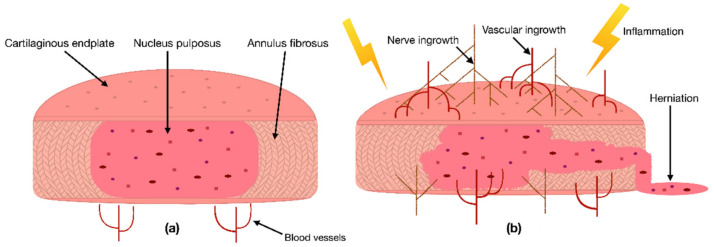
(**a**) Scheme of a cross-section of a healthy intervertebral disc (IVD) with its three tissue types: the nucleus pulposus (NP) in the center, surrounded by the annulus fibrosus and enclosed by two cartilaginous endplates. (**b**) Degenerated inflamed IVD characterized by nerve ingrowth, vascular ingrowth, reduced height, and herniation of the NP through the AF.

**Figure 2 ijms-22-03519-f002:**
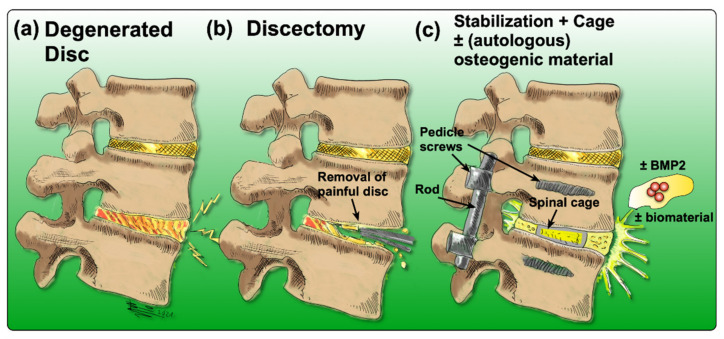
Drawing of the process of achieving spinal fusion: (**a**) the degenerated disc diagnosed by means of imaging, (**b**) discectomy of the degenerated disc, (**c**) stabilization of the spine by insertion of a cage (to maintain the disc height as a spacer) and the addition of an osteoconductive and/or osteoinductive filler material; optionally, BMP2 may be added, coupled with or without a biomaterial.

**Figure 3 ijms-22-03519-f003:**
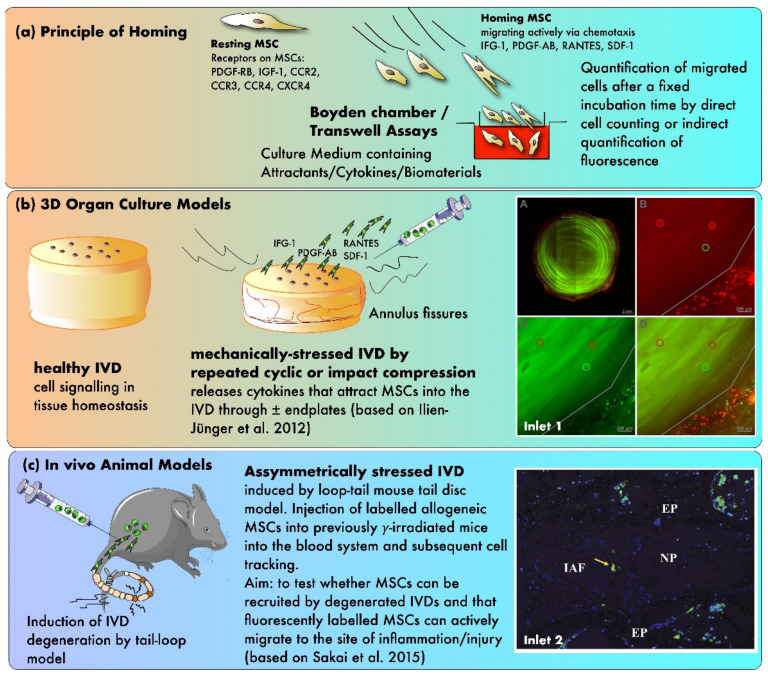
The principles of testing MSC homing in mechanically and/or nutritionally stressed IVDs. (**a**) Principle of homing using in vitro Boyden chamber assays (**b**) Testing of homing in 3D organ culture models (**c**) Evidence from in vivo animal models. Inlet 1: confocal laser scanning pictures (cLSM) of MSC homing experiment with differently labeled MSCs (green, red, and yellow) that were added at various time points during the experiment (A) a transverse section of a bovine IVD without endplates (B–D) close-up pictures at the outer AF (based on Illien-Jünger et al. [93]) Inlet 2: fluorescent image illustrating single stained MSCs in the outer periphery of the AF after 12 weeks post-injection, 10× magnification (based on Sakai et al. [92]). Inlet 2 was reproduced with copyright approval from the publisher.

**Table 1 ijms-22-03519-t001:** Overview of reviews on preclinical studies about the application of MSCs for IVD regeneration published in the past five years. PubMed search with following query box: (((MSC) OR (mesenchymal stem cell)) OR (mesenchymal stromal cell)) AND ((intervertebral disc repair) OR (intervertebral disc regeneration)) Filters: Review, Systematic Review, in the last 5 years. Reviews with a focus on clinical trials were excluded. The search on PubMed was conducted on January 14, 2021.

Study Design	Outcomes	References
Reviewing current uses and potential applications of MSCs in orthopedic surgery.	MSCs can be used for treating musculoskeletal diseases. Further research is needed to evaluate the safety and effectiveness of MSC treatment in orthopedics.	[50]
Reviewing current cell-based therapies for treating IDD, with an emphasis on endogenous repair strategies.	Intradiscal cell injections show promising results to reduce LBP. Endogenous repair with growth factors and chemokines has the potential to overcome hurdles of cell-based therapies.	[36]
Reviewing current knowledge about IDD and discussing recent advancements made with the GDF family for IVD regeneration.	GDF family members can stimulate anabolic processes when delivered to NP cells and promote NP-like differentiation when delivered to MSCs.	[51]
Reviewing characteristics and potency of progenitor cells in different IVD compartments.	IVD progenitor cells show a trilineage differentiation potential and express typical MSC markers. Aging and a degenerated microenvironment affect the fate of IVD progenitor cells.	[52]
Reviewing the successes, drawbacks, and the failures of stem cell-based regenerative medicine approaches to repair IDD.	MSC-based treatments for IDD are on the rise and many of them look promising. Nevertheless, it remains important to understand the fate and contribution of these cells and consequently to promote a safer outcome for stem cell-based approaches.	[53]
Researchers and clinicians discuss the pros and cons of MSC treatment for IVD regeneration.	Preclinical trials using MSCs for IVD regeneration look promising because of MSC proliferation characteristics, anabolic functionality and inflammation-modulatory properties.	[54]
Reviewing mechanisms of endogenous repair during IDD.	Endogenous stem/progenitor cell-based therapy is a promising approach for IDD. Biomimetic peptide biomaterials with signaling molecules can be designed to facilitate the survival and migration of IVD stem/progenitor cells.	[55]
Reviewing strategies for IVD repair using bioscaffolds and MSCs.	Preclinical studies with ovine and canine MSCs show impressive results for IVD repair. The authors also hypothesize that combined therapeutic approaches using biomaterial and cell-based therapies promise notable breakthroughs in IVD repair in the near future.	[56]
Reviewing the therapeutic potential of MSC-derived and IVD-derived extracellular vesicles for IDD.	MSC-derived extracellular vesicles promote ECM synthesis, IVD cell proliferation, and reduce inflammation and apoptosis.	[57]
Reviewing stem cell-based treatments, the molecular machinery and signaling pathways responsible for cartilage and IVD regeneration.	MSC-based therapies show a significant potential to revolutionize the treatment of cartilage defects and IDD. However, there are still many hurdles associated with isolating, expanding, differentiating, and preconditioning MSCs for transplantation into degenerated joints and IVDs.	[58]
Reviewing current stem cell therapies to treat discogenic LBP.	Preliminary animal models have shown the great potential of MSC implantation in order to restore the ECM and regenerate the IVD.	[59]
Reviewing different stem cell-based treatments for IDD.	The transplantation of adult stem cells has repeatedly shown to help regenerate the IVD’s ECM. However, the efficacy of adult stem cell transplantation for IDD treatment is still unclear and therefore needs further investigation.	[60]
Reviewing different stem cell types used as a cell-based therapy for IVD regeneration.	Adult stem cell therapy shows promise for the treatment of IDD. Recent studies have demonstrated the effectiveness of autologous MSC transplantation for IVD regeneration in reproducible animal models.	[61]
Reviewing characteristics of healthy and degenerated IVD microenvironments and their influence on IVD and MSC biological activity and viability.	IDD causes an aggravation of the hostile microenvironment for tissue repair and cell survival in the IVD. However, intradiscal cell therapy with MSCs has the potential to regenerate the IVD and to reverse the changes of IDD.	[41]
Reviewing the latest advances in repairing degenerated IVDs using MSCs, pluripotent stem cells, and NP progenitor cells.	Various animal models have shown that intradiscally transplanted MSCs generally fail to survive and engraft into the IVD niche, whereas pluripotent stem cells and NP progenitor cells can survive successfully.	[62]

Abbreviations: IVD: intervertebral disc, MSC: mesenchymal stromal cell, IDD: intervertebral disc degeneration, LBP: low back pain, GDF: growth and differentiation factor, NP: nucleus pulposus, AF: annulus fibrosus, PAX: paired box, SHH: sonic hedgehog signaling molecule, SOX: SRY-Box transcription factor, FOXA: forkhead box, and ECM: extracellular matrix.

**Table 2 ijms-22-03519-t002:** Published clinical studies related to intradiscal transplantation of MSCs.

Study	Inclusion Criteria	Number of Patients	Number of Cells Injected	Follow Up	Results	References
Injection of autologous BM-derived MSCs into the IVD.	(1) IDD with posterior IVD bulge, (2) radicular pain, (3) failed conservative treatment, (4) failed interventional therapy, (5) patient refuses to pursue surgical option	33	N/A	6 years	Three patients reported. No serious adverse events. Improved SANE numeric pain score. 85% of patients showed reduced IVD bulge size.	[72]
Injection of autologous stromal vascular fraction containing adipose tissue-derived MSCs together with platelet rich plasma.	(1) Between 19 and 90 years of age, (2) LBP after failed conservative treatment for 6 months, (3) fibrous ring able to hold the cell implantation	15	30–60 × 10^6^	6–12 months	Significant improvement in flexion, VAS, PPI, and pain. Positive trends for ODI and BDI. No severe adverse events were observed.	[73]
Injection of autologous BM-derived cultured in a hypoxic environment.	(1) Between 18 and 65 years of age, (2) IDD and failed conservative treatment, (3) significantly functional disability due to pain, (4) painful annular fissures and low pressure positive discography	5	15.1–51.6 × 10^6^	4–6 years	No adverse events were reported. Improvement in mobility, strength, and post-stem cell treatment.	[74]
Injection of adipose tissue-derived MSCs combined with hyaluronic acid derivates.	(1) Between 10 and 70 years of age, (2) LBP for at least 3 months, (3) VAS ≥ 4, (4) ODI ≥ 30, (5) Pfirrmann’s grade III–IV, (6) IDD confirmed by discography	10	20 × 10^6^ (n = 5) and 40 × 10^6^ (n = 5)	12 months	No adverse events were observed. Improvement in VAS and ODI. Elevated IVD water content in three patients.	[75]
Injection of allogeneic BM-derived MSCs into the IVD compared to sham injection.	(1) IDD and remaining LBP after conservative treatment >6 months, (2) fibrous ring able to hold the cell implantation, (3) decrease in disc height >20%, (4) no spinal infection, (5) absent pregnancy in fertile women	24	25 × 10^6^	12 months	Procedure was feasible and safe. Improved algofunctional indices and Pfirrmann’s grade with MSC-treated patients.	[71]
Injection of autologous BM-derived MSCs into the IVD.	(1) Centralized chronic LBP for ≥6 months, (2) non-operative treatment for 3 months without resolution, (3) Pfirrmann’s grade 4–7, (4) Modic grade II change or less, (5) decrease in disc height <30%, (6) ODI ≥ 30/100 (7) VAS ≥ 4/10	26	5426 CFU-F	3 years	Improvement in VAS and ODI. 40% showed improvement on Pfirrmann’s grade despite the relatively low number of CFU-F.	[76]
Injection of autologous BM-derived MSCs into the IVD.	(1) Decrease in disc height >50%, (2) no spinal infection, (3) stages 2, 3, and 4 of Adams, (4) LBP with IDD of one or two IVDs after conservative treatment for over 6 months, (5) No spinal infection	10	10 ± 5 × 10^6^	12 months	85% of pain and disability improvement. Elevated water content but no height recovery in IVDs.	[70]
Transplantation of a collagen sponge containing autologous BM-derived MSCs into the IVD.	(1) IDD confirmed with MRI, (2) vacuum phenomenon, (3) IVD instability, (4) pressure and spontaneous pain at level of degenerated IVD, (5) failed conservative treatment	2	N/A	2 years	Enhanced pain scores and increased water content in the IVD.	[39]

Abbreviations: IVD: intervertebral disc, MSC: mesenchymal stromal cell, BM: bone marrow, LBP: low back pain, IDD: intervertebral disc degeneration, VAS: visual analogue scale, ODI: Oswestry Disability Index, PPI: present pain intensity, BDI: Beck Depression Inventor, CFU-F: colony forming unit -fibroblasts, SANE: Single Assessment Numeric Evaluation, and MRI: magnetic resonance imaging.

**Table 3 ijms-22-03519-t003:** Recent studies related to the homing capabilities of MSCs into the IVD.

Species	Study Type	Cell Types	Outcomes	References
Human	in vitro	BM MSCs	Growth factors and chemokines such as IGF-1, PDGF-AB, RANTES, and SDF-1 showed a chemoattractive effect on MSCs.	[95]
Bovine IVDs and human MSCs	ex vivo	BM MSCs	An intradiscal injectable hydrogel-based on hyaluronan-poly(N-isopropylacrylamide) and supplemented with SDF-1 showed a chemoattractive effect on MSCs.	[97]
Bovine IVDs and human MSCs	ex vivo	BM MSCs	The concentration of RANTES was significantly elevated in the medium of induced degenerated IVDs; RANTES may be a key chemoattractant for MSCs in the IVD.	[98]
Bovine IVDs and human MSCs	ex vivo	BM MSCs	MSC subpopulations positive for CD146 were associated with a greater homing potential but produced a weaker regenerative response than CD146-negative MSCs.	[99]
Murine model with human MPSCs	in vivo	Umbilical cord blood MPSCs	Intravenously injected MSCs showed limited ability to home into a degenerated IVD, but they upregulated GAG and ACAN.	[100]
Murine	in vivo	BM MSCs	Intravenously injected MSCs significantly decreased IVD herniation and induced an immunomodulatory effect.	[101]
Murine	in vivo	BM MSCs	Only a limited number of intravenously injected MSCs migrated to a degenerated IVD. However, the more serious the injury, the more cells were recruited.	[92]
Bovine IVDs and human MSCs	ex vivo	BM MSCs	Greater MSC homing occurred with degenerated IVDs than healthy samples, and IGF-1-transduced MSCs significantly increased the proteoglycan synthesis.	[93]
Bovine IVDs and human MSCs	ex vivo	BM MSCs	MSCs seeded on the endplate’s surface of nucleotomized IVDs migrated into the NP and stimulated ECM production and growth factors.	[104]
Bovine and human	ex vivo	BM MSCs	Homed MSCs increased the fraction of Tie2-positive IVD cells, enhanced IVD cell proliferation, and reduced the fraction of dead cells in the IVD.	[105]

Abbreviations: IVD = intervertebral disc, BM = bone marrow, MSC = mesenchymal stromal cells, MPSC = multipotent stem cells, IGF-1 = insulin-like growth factor 1, PDGF-AB = platelet-derived growth factor -AB, SDF-1 = stromal cell-derived factor 1, CD146 = cluster of differentiation 146, GAG = glycosaminoglycan, ACAN = aggrecan, ECM = extracellular matrix, and Tie2 = angiopoietin-1 receptor.

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
