# Peer review of "The Application of Mesenchymal Stromal Cells and Their Homing Capabilities to Regenerate the Intervertebral Disc"

_ijms, 2021, doi:10.3390/ijms22073519_

Round 1

Reviewer 1 Report

The present manuscript titled "The Application of Mesenchymal Stromal Cells and their Homing Capabilities to Regenerate the Intervertebral Disc" gives a review of literature of about the use of MSCs in the treatment of degenerated intervertebral disc for the treatment of lower back pain. The manuscript has done a thorough review of literature and is written well. The organization is also done well. There are a few minor points that will further improve the manuscript-

  1. The schematic in Figure 1 needs to be more representative of the IVD anatomy. It can be improved on.
  2. Table 1 will benefit from reorganization. Instead of the article name in the cell type column, it would be good to describe how they have used MSCs and another column describing the outcomes will add more information to the table. There is no need of author name and year of publication as it is already cited in the reference, more information from these articles would be better. 

 Author Response

Reviewer 1

Comments and Suggestions for Authors

The present manuscript titled "The Application of Mesenchymal Stromal Cells and their Homing Capabilities to Regenerate the Intervertebral Disc" gives a review of literature of about the use of MSCs in the treatment of degenerated intervertebral disc for the treatment of lower back pain. The manuscript has done a thorough review of literature and is written well. The organization is also done well. There are a few minor points that will further improve the manuscript-

  1. The schematic in Figure 1 needs to be more representative of the IVD anatomy. It can be improved on.

Answer: We thank the reviewer for this favorable feedback. We agree with the reviewer that Figure 1 should be more representative of the IVD anatomy. For this reason, we made several improvements to it. This includes an enlargement of the nucleus pulposus (NP), the addition of more concentric rings or lamellae in the annulus fibrosus (AF), alternating fibers in the AF, thickening of the cartilaginous endplates (CEP), adding capillary buds to the CEP, and adding non-penetrating blood vessels to the healthy IVD.

  1. Table 1 will benefit from reorganization. Instead of the article name in the cell type column, it would be good to describe how they have used MSCs and another column describing the outcomes will add more information to the table. There is no need of author name and year of publication as it is already cited in the reference, more information from these articles would be better. 

Answer: This is a very valid point. We have taken the reviewer’s suggestions into consideration and revised Table 1 accordingly. Instead of the article name, author name and year of publication, the table now contains the study design and the general outcomes of the studies listed. Additionally, the author name and year of publication has also been deleted from Table 2 to ensure constancy throughout the review.

Reviewer 2 Report

Authors are advised to include a paragraph accompanied by an illustration that briefly explains what the spinal fusion procedure consists of.

Author Response

Reviewer 2

Authors are advised to include a paragraph accompanied by an illustration that briefly explains what the spinal fusion procedure consists of.

Answer: We thank the reviewer for highlighting the need to include an illustration and expanding on the paragraph about the procedure of spinal fusion. In the revised version, the procedure of spinal fusion is now explained in more detail. Furthermore, the section is now accompanied by an additional figure (Figure 2). The section about spinal fusion now reads as follows (page 3, line 98-103):

“In brief, the degenerated IVD is removed, and the cavity between the adjacent vertebral bodies is replaced by a cage containing bone grafts or substitutes that can be supplemented with osteogenic inductive growth factors like the bone morphogenetic protein 2 (BMP2) to induce ossification. Then pedicle screws are inserted to improve the mechanical stability and to immobilize the adjacent vertebral bodies (Figure 2).